# The Influence of Message Framing on Elderly Tourists’ Purchase Intentions of Health Services: A Case Study of Guangxi Bama

**DOI:** 10.3390/ijerph192114100

**Published:** 2022-10-28

**Authors:** Ji Wen, Xiaolin Mai, Wei Li, Xin Liu

**Affiliations:** 1Shenzhen Tourism College, Jinan University, Shenzhen 518053, China; 2School of Management, Jinan University, Guangzhou 510632, China

**Keywords:** elderly tourists, message framing, perceived benefits, perceived susceptibility, purchase intention

## Abstract

Traveling has become an increasingly important lifestyle for the elderly to realize active aging. The elderly are more inclined to pay attention to wellbeing-related products whilst on vacations, representing a market opportunity for providers of tourism health services. This study conducted an experiment to explore how message framing affects intentions to purchase health services in elderly tourists over the age of 59 years. A total of 216 elderly tourists from Bama, a famous wellness tourism destination in China, were recruited as participants for a single-factor (message framing: positive versus negative) experiment. Our results indicate the following: (1) message framing has a significant impact on elderly tourists’ intentions to purchase health services. Compared with messages that are negatively framed, positive messages are more persuasive. (2) Perceived benefits mediate the above relationship. (3) Perceived susceptibility moderates the impact of message framing around perceived benefits, as well as the indirect effect of perceived benefits on purchase intention. Theoretically, this paper clarifies the mechanism and conditions of message framing in relation to its effects on consumer intentions, enriching knowledge about the intersection between tourism and health consumption. This paper also provides guidance for providers of health tourism who are seeking to capture the market of elderly tourists.

## 1. Introduction

Population aging is one of the major challenges facing the world today. In 2021, there were more than 1.08 billion people aged 60 and above, accounting for 13.7% of the global population [1]. In an attempt to address the problem of population aging in the 21st century, the World Health Organization put forward the concept of “active aging”, which refers to the process of creating optimal health, social participation opportunities, and safety for the elderly to improve their quality of life [2]. Existing research indicates that traveling is one of the most effective ways to realize active aging, since traveling provides opportunities for older people to communicate with each other [3], and it promotes their physical and mental health [4,5] and improves their quality of life [6,7]. Compared with tourists of other ages, elderly tourists tend to be more motivated to engage in activities that benefit their physical and mental health [8]. They tend to be drawn to activities involving nature, learning about other cultures, and activities promoting wellbeing [9]. Indeed, the provision of health and wellness services for the elderly in tourism has been a growing research field. Health services refer to healthcare within disease prevention, medical treatment and rehabilitation [10]. Many of the elderly today have a medical condition of some sort, often requiring regular medications or health-related services [11,12]. By purchasing and using health services, the elderly can alleviate their discomfort to a certain extent. They can meet their own health needs and obtain a good health travel experience. At present, research on elderly tourists has focused on their motivations, decision-making, and behavioral characteristics [13,14,15,16,17]. There has been limited research on the consumption patterns and decisions of elderly tourists with respect to health and wellness services, a knowledge gap this paper seeks to address. Against a background of the severe aging situation of the population, it is very important to explore the antecedents and boundary conditions of elderly tourists’ willingness to purchase health services, which are of certain significance to promote the realization of active aging.

The pursuit of health and wellness services by elderly tourists may be understood as an attempt by such individuals to promote and self-manage their own health and to realize the goal of active aging [18,19]. The information that drives this pursuit of health and wellness services, however, is never objective and usually deliberately framed in ways to drive particular consumer behaviors [20,21,22]. When individuals process any piece of information, there is an intermediary path between the information stimulus and the resulting behavior of that individual, which involves perceptions or feelings on the part of the individual [23,24]. Consumer purchasing decisions are primarily driven by the individual perceiving the benefits to be gained from consuming the product [25,26,27] and, indeed, perceptions of the benefits that will eventuate often promote health-related behaviors [28]. Therefore, this study speculates that perceived benefits may be a mediating variable between message framing and intentions to purchase health services, explaining how information about health and wellness products and services drive health tourism consumption behaviors. Specifically, due to the different focus of information expression, elderly tourists may have different subjective perceptions of the benefits towards the same health service, which may eventually affect the possibility of purchasing this service. Moreover, the persuasiveness of information presentation is influenced by personal characteristics [29]. Previous studies have shown that some individuals may be especially susceptible to particular ways in which messages are framed. For example, if an individual is anxious about their health, messages about the dangers of neglecting one’s health may be more amplified [30,31,32]. This study focuses on susceptibility as a boundary condition of the message framework in the context of tourism health consumption for further discussion.

The following research questions drive this paper: (1) does the framing of a message significantly affect the willingness of elderly tourists to purchase health services? (2) Do elderly tourists’ perceptions of the benefits to be gained constitute the mediating mechanism of the main effect? Finally, (3) how does one’s susceptibility mediate the relationship between the way a message is framed and one’s purchasing intentions? This study draws on socioemotional selectivity theory [33] and framing theory [34], which respectively provide important insights for understanding the cognitive processing of the elderly and how different expressions of information presentation affect individuals’ decision-making, as well as relevant literature to propose a moderated mediation model. It focuses on the intersection of senior tourism and health behaviors. It also offers some practical guidance for health tourism providers who are looking to capture the elderly market.

## 2. Literature Review and Hypothesis Development

### 2.1. The Positivity Effect of the Aged

Research has indicated that, by and large, the elderly prefer positive messaging when cognitively processing information compared to negative frames of reference [35]. This is connected to the cognitive and emotional characteristics of the elderly whose emotional functions remain high even as their cognitive abilities decline [36]. Socioemotional selectivity theory (SST) can help to explain the purchasing intentions and decisions of elderly tourists. SST holds that as individuals grow older, they become increasingly aware of their mortality and the limited time they have left. This leads to a shift in priorities where knowledge acquisition is no longer the main goal and is superseded by the pursuit of emotional satisfaction [33,37,38,39]. This motivational change toward emotional regulation directs cognition resources [40,41]. Thus, the avoidance of negative emotions is common in the elderly as these are threatening to the objective of deriving meaning and satisfaction from one’s life [42] and the inclination towards positive emotions and messages is known as the positivity effect.

In rudimentary terms, the “positivity effect” in the elderly works as follows. To improve and maintain one’s positive emotional experiences, elderly people show a preference for positive messages in attention and memory, and reduce the processing of negative information, which in turn affects their behavioral decision-making. Existing research has shown that, in contrast to negative information, positive pictures, such as happy faces, are more likely to attract the attention of the elderly [43,44] and leave a deeper impression on them [45]. Presenting information about a project in a positive way is more likely to stimulate their enthusiasm and willingness to participate [46]. The positivity effect is used in this study to analyze the relationship between how messages are framed and the purchasing intentions and decisions of elderly tourists.

### 2.2. Message Framing and Purchasing Intentions

Purchasing intentions (PI) involve the possibility of consumers buying a certain product or service [47,48]. We use this term to refer to the intentions of elderly tourists to purchase health-related services. Message framing refers to the way information is presented. Changing the manner of information expression can affect individuals’ decision preferences and result in framing effects [34]. Messages can be framed in positive or negative ways: positive messages focus on the benefits, positive attributes, and results of a certain thing or behavior, while negative messages emphasize risks, negative attributes, and results [49,50]. As discussed previously, the purchasing of health services by elderly tourists is also an effort to promote one’s health. Messages around health services and products tend to emphasize the specific consequences of one’s actions or inaction [51,52]. Where positive framing emphasizes the positive outcomes of buying and using health services, negative framing emphasizes the negative consequences of not using the health services on offer [53].

Existing studies have shown that messaging framing significantly affects individual behavioral responses [20,22,54,55]. Research around health behavior indicates that information framed around potential gains is more likely to promote disease prevention behavior [56], while messages framed around potential losses are more likely to convince individuals to perform actions that help with the detection of diseases [51]. As previously noted, information framed in a positive way is more likely to motivate the elderly to adopt healthy behaviors, such as participating in physical activities [46,57]. Messages which describe the positive effects of health services can address elderly preferences for emotional satisfaction and align with the “positivity effects” of their cognitive processing. This leads to our first hypothesis.

**Hypothesis** **1** **(H1).***Compared to negative framing, positive framing leads to stronger purchasing intentions in the elderly*.

### 2.3. The Meditating Role of Perceived Benefits

Perceived benefit (PB) is the benefit that individuals perceive they will derive from a product or service. PB is one of the key factors that determine the purchasing decisions of consumers [25,26,27]. In the field of health communication, PB refers to an individual’s perception and evaluation of the potential benefits and positive effects that result from a health-related behavior, and it reflects the individual’s belief about the effectiveness of that behavior in promoting one’s health [28]. PB is used in this study to refer to the subjective feelings of elderly tourists with respect to the benefits of purchasing health services to promote their health. PB is an important predictor of individual consumption decisions and healthy behaviors, because the higher the perceived benefits of individuals, the stronger their willingness to buy products and services [58,59,60,61], and the greater the possibility they will carry out healthy behaviors [62,63].

We propose that PB plays a mediating role in the relationship between message framing and the behavioral responses of elderly tourists. That is, positive messages affect PB and influences people to adopt healthy behaviors [64]. The information presented is considered positive when it emphasizes the benefits or positive outcomes that people can gain from adopting the behavior [57]. Our next proposition is as follows:

**Hypothesis** **2** **(H2).**
*The relationship between message framing and purchasing intentions is mediated by perceived benefits.*


### 2.4. The Moderating Role of Perceived Susceptibility

Perceived susceptibility (PS) is an individual’s subjective perception of the likelihood that he or she will suffer from a certain disease or fall into a state of illness [31]. In our study, PS involves the extent to which elderly tourists believe that they may be at risk of experiencing ill health. As a health belief, perceived susceptibility can significantly influence an individual’s health-related attitudes and behaviors [28,65,66]. Individuals vary widely in their perceptions of their susceptibility to diseases and ill health [67]. Personality factors and individual dispositions affect an individual’s susceptibility to health problems [68] and perceived susceptibility is a variable related to individual differences [69].

We contend that individual differences will affect the effects of different message strategies [29]. If individuals have high levels of PS and think they are vulnerable to certain diseases or conditions, they may be more attentive to health-related messages [30,31,32,69] than those who have low levels of PS. This leads to our next hypothesis.

**Hypothesis** **3** **(H3).**
*Perceived susceptibility moderates the relationship between message framing and purchasing intentions, such that the relationship is stronger when perceived susceptibility is high.*


To sum up, this study draws on socioemotional selectivity theory and framing theory to suggest that the indirect path between message framing and purchasing intentions is positively moderated by perceived susceptibility. That is, when perceived susceptibility is high, the indirect effect of perceived benefits is stronger and vice versa. We offer the following moderated mediation model (see Figure 1).

**Hypothesis** **4** **(H4).**
*Perceived susceptibility moderates the indirect effect between message framing and purchasing intentions via perceived benefits, such that the indirect effect is stronger when perceived susceptibility is high.*


## 3. Experimental Design

### 3.1. Research Context

This study was conducted in Bama, Guangxi, which is one of the most famous wellness tourism destinations in China. Bama attracts more than 3,000,000 health tourists every year due to the quality of its natural environment and its reputation for longevity, leading many researchers to focus on this location as the research context of wellness tourism [9,70,71]. Bama’s visitors tend to be highly concerned about their overall health, many suffering from chronic diseases [9], and are drawn to its local healthcare facilities [72,73], most of which offer traditional Chinese medicine (TCM) healthcare. TCM-related activities are an important form of wellness tourism in China [70]. TCM providers claim that they can help people to relieve their discomfort and relax their mind and body [74]. As a unique health resource, TCM is an indispensable part of health consumption in the elderly in China [75], most of whom believe it improves their quality of life [76]. TCM is believed to play an important role in preventing diseases, delaying the occurrence of diseases, and promoting the recovery of diseases. It is common in daily life [77] and includes acupuncture, massage, cupping, scraping, fumigation and washing, pasting, plasters, and foot therapy.

### 3.2. Questionnaire Design

#### 3.2.1. Stimuli Materials

To determine the persuasive effects of the given messages [78,79], this study established a control group who were exposed to messages that were neither positively nor negatively framed. A second group was exposed to messages that were positively framed and a third to negatively framed messages. In line with previous studies [32,80], the experimental materials for each group were compiled as follows.

Positive Framing: TCM healthcare is a TCM technique that treats the human body under the guidance of TCM theory. It includes acupuncture, moxibustion, cupping, scraping, massage, drug fumigation, acupoint application therapy, foot therapy, etc. If you use TCM healthcare, you might gain some favorable health-related benefits, such as (a) reducing the possibility of senile health problems and preventing the occurrence and development of diseases, (b) early detection of disease symptoms, timely treatment, and a reduction in complications, and (c) promoting rehabilitation of senile and chronic diseases.

Negative Framing: TCM healthcare is a TCM technique that treats the human body under the guidance of TCM theory. It includes acupuncture, moxibustion, cupping, scraping, massage, drug fumigation, acupoint application therapy, foot therapy, etc. If you do not use TCM healthcare, you might face some unfavorable health-related outcomes, such as (a) increasing the possibility of senile health problems and causing the occurrence and development of diseases, (b) late detection of disease symptoms, untimely treatment, and an increase in complications, and (c) hindering rehabilitation of senile and chronic diseases.

No Framing: TCM healthcare is a TCM technique that the treats human body under the guidance of TCM theory. It includes acupuncture, moxibustion, cupping, scraping, massage, drug fumigation, acupoint application therapy, foot therapy, etc.

#### 3.2.2. Variable Measurement

The measurements of the main variables of purchasing intentions, perceived benefits and perceived susceptibility were, respectively, based on the works of Han et al. [81], Wang and Li [61] and Champion [82]. Each measurement of variables consists of three items and all are measured by a 5-point Likert scale (1 = “strongly disagree”, 5 = “strongly agree”). To reduce subjective deviation, this study adopted a translation and back-translation method and made adjustments according to the reading habits of the elderly and the feedback of researchers. To confirm the validity of the manipulation, this study used Meyers-Levy and Maheswaran’s [83] method for reference. After reading the material, participants were asked to judge the content of the material by choosing the most appropriate description among the following three options: “Advantages of using TCM healthcare”, “Disadvantages of not using TCM healthcare”, and “Introduction of TCM healthcare only”. In addition, demographic variables of the elderly (gender, age, education, monthly income, illness, self-perceived health status, demand for TCM healthcare) were used as control variables to exclude the interference of unrelated factors.

### 3.3. Pretest

To test the effectiveness of the manipulation of independent variables and the facticity and understandability of the materials, this study conducted a pre-test. A total of 60 participants were recruited through an online questionnaire platform, and all of them were randomly assigned to each experimental group. Seventeen of the 20 subjects in the no-framing group chose “Introduction of TCM healthcare only” (χ^2^ = 43.333, df = 2, *p* < 0.05), 18 of the 20 subjects in the positive framing group chose “Advantages of using TCM healthcare” (χ^2^ = 28.937, df = 2, *p* < 0.05), and 16 of the 20 subjects in the negative framing group chose “Disadvantages of not using TCM healthcare” (χ^2^ = 39.562, df = 2, *p* < 0.05). The results of our chi-square test showed that the subjects were able to distinguish between the different message frames, confirming that the manipulation of the independent variables was successful. In addition, from the perspective of elderly tourists, the experimental materials are quite in line with the actual daily situation (M _Facticity_ = 4.07, SD _Facticity_ = 0.800), and the content is easy to understand (M _Understandability_ = 4.38, SD _Understandability_ = 0.846). There was no significant difference in participants’ assessment of facticity [F(2,57) =2.491, *p* = 0.092] and understandability [F(2,57) = 0.645, *p* = 0.529] among the three groups.

The scales used in this study have good reliability and validity. Cronbach’s α of each scale was greater than 0.8 (Table 1), indicating that the measurements had good stability and consistency. Our confirmatory factor analysis was conducted by AMOS, which indicated a good fit of the measurement model (χ^2^/df = 2.367, RMSEA = 0.071 < 0.08, GFI = 0.956 > 0.9, AGFI = 0.917 > 0.9, NFI = 0.969 > 0.9). In addition, factor loadings and average variance extraction (AVE) were greater than 0.5, and combined reliability (CR) was greater than 0.7, indicating that the scale had good convergent validity. Meanwhile, although the variables were significantly correlated, the correlation coefficients were all smaller than the AVE square root of each variable (Table 2), indicating that the scales had good discriminant validity.

### 3.4. Data Collection

The inclusion criteria for this study’s sample were people born between 1946 and 1963. The post-war “baby boomers” have attracted widespread attention in elderly tourism studies at home and abroad, due to their large numbers, the good economic conditions they have experienced, and their keenness to travel [84,85]. This group is also the main target market of wellness tourism [86]. According to the National Bureau of Statistics of China [87], China’s post-war “baby boom” appeared from 1946 to 1963. Now that this generation is reaching the legal retirement age, they are becoming the main population of the elderly tourism market.

This study adopted a field experiment in Bama, Guangxi. We face-to-face recruited elderly tourists visiting local tourist attractions as participants, and they were invited to take part in the experiment in a real environment. This approach served to improve the external validity of our data [88]. Meanwhile, we also recruited participants through an online questionnaire platform (www.sojump.com, accessed on 29 March 2022) and the WeChat group chats of local commonweal organizations. In both online and offline experiments, three groups of experimental questionnaires were shuffled and randomly distributed to subjects, and each participant had access to one of the three experimental conditions. A total of 316 questionnaires were distributed (173 online and 143 offline), and 216 of them were valid. The effective rate of questionnaire was 85.1%.

The demographic characteristics of the sample are shown in Table 3. The proportion of elderly women is higher than that of elderly men. It conforms to the characteristics of aging populations in China and across the world [1]. Our sample was mainly composed of those who fell in the 62 to 71 years (60.9%) age group and 73.6% of our sample had completed at least high school education and were receiving more than 2000 yuan every month (71%). Although many in our sample were suffering from a condition of some kind, their overall health status was above the medium level. To sum up, our participants, most of whom had a reasonable level of education, time, money, and health, were in a position to travel.

## 4. Results

Before hypothesis testing, the chi-square test was used to control the distribution differences of variables in each group. After a pairwise comparison, we found no significant difference in the demographic variables between the three groups (*p* > 0.05), indicating that the experimental control effect was good.

This study used one-way analysis of variance, hierarchical regression, and path analysis methods to verify the main effect, mediating effect, and moderated mediating effect.

First of all, the main effect was tested by ANOVA (Table 4). The results show that different message framings have significant differences in their influence on the purchasing intentions of elderly tourists [F(2, 266) = 10.248, *p* < 0.05)]. The purchasing intentions of those in the positive framing group was significantly higher than that in the negative framing group and the no framing group (M _Positive_ − M _Negative_ = 0.467, M _Positive_ − M _No_ = 0.564, *p* < 0.05, Table 5). Thus, H1 is supported.

Next, this study encoded the framed messages (positive framing = 1, negative framing = 0) and used a hierarchical regression analysis to test the mediating effect. The results are shown in Table 6. After controlling the demographic variables, we found that message framing has a significant positive impact on intentions to consume TCM healthcare (M5, β = 0.240, *p* < 0.001) and on perceptions of its benefits (M2, β = 0.216, *p* < 0.001). Perceived benefits have a significant positive impact on purchasing intentions with respect to TCM healthcare (M6, β = 0.753, *p* < 0.001). Further, SPSS-PROCESS Macro was adopted [89], and model 4 (mediating model test) was selected, with 5000 as the set number of bootstrapping. Our results show that the mediating effect coefficient of perceived benefits was 0.292, and the 95% confidence interval was (0.111, 0.489), excluding 0. This indicates that the mediating effect of perceived benefits is significant, supporting H2.

To test the moderating role of perceived susceptibility on the relationship between message framing and perceived benefits (H3), a hierarchical regression was conducted again. This study decentralized perceived susceptibility and constructed an interaction variable between message framing and perceived susceptibility to exclude the influence of multicollinearity. As shown in Table 6, the regression coefficient of the interaction term between message framing and perceived susceptibility was significant (M3, β = 0.177, *p* < 0.05). This indicates that perceived susceptibility has a significant moderating effect on the relationship between message framing and perceived benefits. To further understand the moderating effect of perceived susceptibility, we conducted a simple slope analysis, and the results are shown in Figure 2. For elderly tourists with high perceived susceptibility (mean + 1SD), there is a significant correlation between message framing and perceived benefits, with an effect coefficient of 0.394, *p* < 0.05. For elderly tourists with low perceived susceptibility (mean − 1SD), there is no significant correlation between message framing and perceived benefits, and the effect coefficient was −0.034, *p* > 0.05. Thus, H3 is supported.

Finally, to further explore and test the moderated mediating effect, the moderated mediation approach [90] was applied to calculate the indirect effect as well as its confidence interval at the high and low levels of the moderator. A bootstrap method [88] was used to test the moderated mediation effect of task interdependence (5000 bootstrap samples, Model 7 in SPSS PROCESS Macro). After controlling the demographic variables, the multiple regression outcomes were obtained (see Table 7). In the high perceived susceptibility condition, the mediation of perceived benefits (B = 0.338, Bootstrap SE = 0.125, LLCI = 0.104, ULCI = 0.590, confidence interval did not contain 0) reached the significant level. However, in the low perceived susceptibility condition, the indirect path did not reach a significant level (B = −0.029, Bootstrap SE = 0.156, LLCI = −0.339, ULCI = 0.266, it did not exclude 0). Thus, H4 is supported.

## 5. Conclusions and Discussions

### 5.1. Conclusions

This study sought to reveal how message framing affects the intentions of elderly Chinese tourists to purchase health services. Drawing on socioemotional selectivity theory (SST), we found that positively framed messages which describe the benefits of health services can significantly motivate the purchasing intentions of elderly tourists, whereas negatively framed messages which outline the risks of not accessing these services have no significant effect. Our results align with previous research that demonstrate the “positivity effect” on cognitive processing in older adults [43,57,91].

Perceived benefits mediate the relationship between message framing and purchasing intentions. That is, the framing of a message can affect the perceptions of elderly tourists in relation to the benefits of purchasing and using health services, which then influences their purchasing intentions. Positive messages can strengthen the belief in elderly tourists that health services are beneficial, prompting them to engage in behaviors that they believe will be conducive to their health.

Perceived susceptibility enhances the impact of message framing on perceived benefits. When elderly tourists perceive themselves to be more likely to get sick, they pay more attention to information relating to health management. As such, messages which are positively framed can have a stronger impact on their perceptions of the benefits to be gained by purchasing the health services that are on offer. In addition, perceived susceptibility also moderates the indirect path strength of message framing–perceived benefit–purchase intention. While elderly tourists with high perceived susceptibility will be more persuaded by positive messages about the benefits of purchasing health services, elderly tourists with low perceived susceptibility are less affected by positive messages since they believe they are in good health and the need for health services is less urgent [92]. In other words, there is no significant difference in the purchasing intentions of those who have low levels of perceived susceptibility when they are exposed to positive and negative messages.

### 5.2. Theoretical Implications

This study represents an intersection between tourism, health, and aging research. In applying SST, one of the main theories in the field of aging (i.e., SST), to the context of consumption behaviors with regards to health tourism, this study demonstrates the relevance of SST to a range of fields other than that of aging.

We also confirmed that the effects of message framing are contingent on differences across individuals [29,93]. As a classical explanation of framing effects, prospect theory assumes that individuals are more sensitive to losses [94]. However, messages describing the negative consequences of not using health services do not significantly increase the purchasing intentions of elderly tourists. Instead, positive messages have a greater persuasive power. SST suggests that, compared with young people, elderly people show a preference for positive information when cognitively processing information [35]. Ultimately, age is also shown to be an important factor in affecting the power of differently framed messages. This study also found that in the context of health tourism, perceived susceptibility is the boundary condition for the effect of message framing. When the perceived susceptibility of elderly tourists is at a high level, positive framing has a stronger impact on their perceived benefits of health services and their purchasing intentions. Identifying the boundary condition helps to explain under what conditions and how to structure messages to give full play to the optimal effect of the information dissemination strategy, so as to provide a new perspective for an in-depth understanding of the effectiveness of the information presentation and dissemination.

### 5.3. Practical Implications

Providers in the health tourism sector would do well to market their services and products in terms of the benefits to be gained when consumers acquire them. Our study has demonstrated that messages which are framed positively have a greater persuasive power among the elderly, compared to messages that are framed negatively. Given that the elderly are a significant segment of the tourism market, providers should look for ways to emphasize the advantages to be gained when their services and products are consumed.

Health tourism providers have much to gain by working with professionals in the health industry to publicize the benefits of the services they offer. An approach that details the benefits that come from consuming health services and adopting healthy behaviors represents the best opportunity for persuading elderly consumers to partake in the health services on offer and to promote their health and wellbeing overall.

Providers of health tourism can also look for opportunities to address the fears of elderly consumers, who are more susceptible to worries about falling ill. Tourism health service enterprises should focus on elderly tourists with high perceptual susceptibility. If health tourism providers worked with practitioners in the health industry to promote key information about preventive measures in relation to common diseases, consumers may feel more assured that the health services they purchase are evidence-based, and not simply profit-driven.

### 5.4. Limitations and Further Research

This study has several limitations, one of which involves our sample group. Although Bama in Guangxi is one of the most popular destinations for elderly tourists in China, the national representativeness of our sample group needs to be further verified. Future research can repeat our experiment by selecting multiple senior wellness tourism destinations, different samples of senior tourists and points in time when the experiment is conducted, so as to improve the universality of our research results. Another limitation involves the textual nature of the messages we used as the stimuli for our participants. Other media, such as images and audio, could also be investigated in the future to determine which forms of information presentation are the most powerful.

## Figures and Tables

**Figure 1 ijerph-19-14100-f001:**
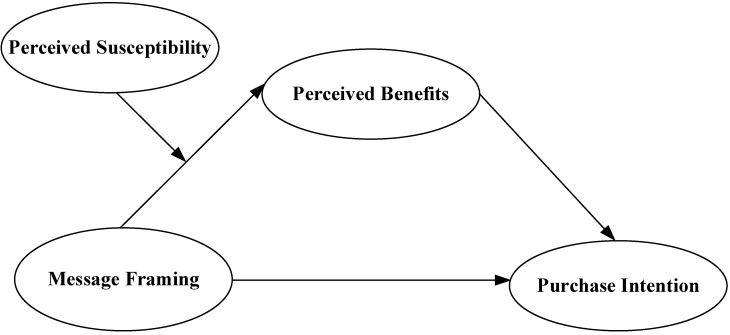
Hypothesis Model.

**Figure 2 ijerph-19-14100-f002:**
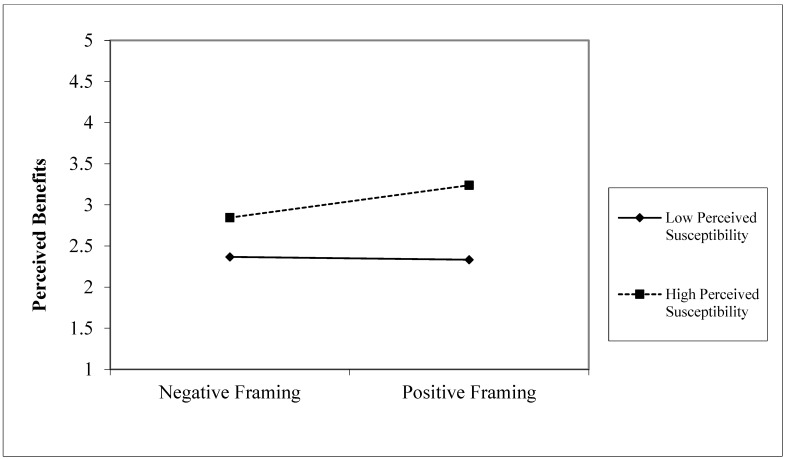
Results of simple slope analysis.

**Table 1 ijerph-19-14100-t001:** Reliability and validity test of variable measurements (N = 269).

Items	Cronbach’s α	Loading	AVE	CR
Purchase Intention	0.919		0.792	0.920
PI1. I would like to buy traditional Chinese medicine services during my trip		0.850 ***		
PI2. I plan to buy traditional Chinese medicine services during my trip		0.909 ***		
PI3. I will try my best to buy traditional Chinese medicine services during my trip		0.910 ***		
Perceived Benefits	0.885		0.729	0.889
PB1. The use of TCM services can reduce physical health problems		0.843 ***		
PB2. The use of TCM services can effectively prevent health problems		0.905 ***		
PB3. The use of TCM services is beneficial for my health		0.810 ***		
Perceived Susceptibility	0.857		0.675	0.861
PS1. My physical condition is likely to lead to health problems		0.874 ***		
PS2. I feel like I am going to have health problems		0.844 ***		
PS3. In the next few years, my condition has a high potential for health problems		0.741 ***		

Note. *** *p* < 0.001.

**Table 2 ijerph-19-14100-t002:** The mean, standard deviation and correlation coefficient of variables (N = 269).

Variable	Mean	SD	1	2	3
1. PI	3.334	0.933	(0.890)		
2. PBs	3.610	0.800	0.821 **	(0.854)	
3. PS	3.363	0.824	0.391 **	0.372 **	(0.822)

Note. ** *p* < 0.01; the numbers in diagonal parentheses denote the square root of AVE for each variable.

**Table 3 ijerph-19-14100-t003:** Demographic Characteristics of the sample (N = 269).

Demographics	Indicators	Frequency	Percentage (%)	Cumulative Percentage (%)
Gender	Male	105	39.0	39.0
	Female	164	61.0	100.0
Age	59–61	72	26.8	26.8
	62–66	87	32.3	59.1
	67–71	77	28.6	87.7
	≥72	33	12.3	100.0
Education	Junior high school and below	71	26.4	26.4
	Senior high school	88	32.7	59.1
	Bachelor’s degree	110	40.9	100.0
	Master’s degree and above	0	0	100.0
Monthly income (RMB yuan)	≤2000	31	11.5	11.5
	2001–4000	109	40.5	52.0
	4001–6000	82	30.5	82.5
	6001–8000	29	10.8	93.3
	≥8000	18	6.7	100.0
TCM demand	No	170	63.2	63.2
	Yes	99	36.8	100.0
Illness	No	40	14.9	14.9
	Yes	229	85.1	100.0
Health Status	Very bad	0	0	0
	Bad	24	8.9	8.9
	Average	107	39.8	48.7
	Well	109	40.5	89.2
	Very well	29	10.8	100.0

**Table 4 ijerph-19-14100-t004:** Results of one-way ANOVA.

DV	IV	Mean	SD	F	Sign
PI	No framing	3.11	0.915	10.248	0.000
	Negative framing	3.20	0.998		
	Positive framing	3.67	0.789		

**Table 5 ijerph-19-14100-t005:** Results of Tukey HSD test after ANOVA.

DV	(I) Material	(J) Material	Difference of Means (I–J)	Standard Error	Sign
PI	No framing	Negative framing	−0.096	0.136	0.760
		Positive framing	−0.564 ***	0.134	0.000
	Negative framing	No framing	0.096	0.136	0.760
		Positive framing	−0.467 **	0.134	0.002
	Positive framing	No framing	0.564 ***	0.134	0.000
		Negative framing	0.467 **	0.134	0.002

Note. ** *p* < 0.01, *** *p* < 0.001.

**Table 6 ijerph-19-14100-t006:** Results of hierarchical regression analysis.

Variable	Perceived Benefits			Purchase Intention			
	M1	M2	M3	M4	M5	M6	M7
**Control Variable**							
Gender	−0.075	−0.090	−0.022	−0.098	−0.114	−0.041	−0.048
Age	−0.032	−0.030	0.034	−0.071	−0.068	−0.047	−0.047
Education	−0.032	−0.066	−0.036	−0.097	−0.136	−0.073	−0.087
Monthly Income	0.072	0.094	0.082	0.150	0.174 *	0.095	0.105
TCM Demand	0.239 **	0.218 **	0.221 ***	0.221 **	0.198 **	0.041	0.038
Illness	0.109	0.097	0.081	0.115	0.100	0.032	0.029
Health Status	−0.151	−0.144	0.019	−0.163 *	−0.155 *	−0.050	−0.050
**Independent Variable**							
Message Framing		0.216 **	0.114		0.240 ***		0.081
**Mediator**							
PBs						0.753 ***	0.734 ***
**Moderator**							
PS			0.303 **				
**Interaction**							
Message Framing × PS			0.177 *				
R^2^	0.149	0.193	0.342	0.178	0.233	0.661	0.667
ΔR^2^	-	0.044	0.016	-	0.055	0.483	0.434
F	4.328 ***	5.154 ***	8.839 ***	5.350 ***	6.514 ***	41.912 ***	38.036 ***

Note. *** *p* < 0.001, ** *p* < 0.01, * *p* < 0.05.

**Table 7 ijerph-19-14100-t007:** Results of the test of moderated mediating effects.

Path	Moderator	Indirect Effects	Standard Error	95% Confidence Interval	
				LLCI	ULCI
**Message Framing × PS→PBs→PI**	M − 1SD	−0.029	0.156	−0.339	0.266
	M	0.155	0.085	−0.012	0.322
	M + 1SD	0.338	0.125	0.104	0.590

## Data Availability

The data presented in this study are available on reasonable request from the corresponding author.

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
