# Peer review of "The Influence of Message Framing on Elderly Tourists’ Purchase Intentions of Health Services: A Case Study of Guangxi Bama"

_ijerph, 2022, doi:10.3390/ijerph192114100_

Round 1

Author Response

Reviewer #1:

Comment 1:This manuscript examines the impact of message framing, perceived benefits, perceived susceptibility on elderly tourists’ purchase intentions of health services. It reports data collected in 216 participants. All of the four research hypotheses are supported. The contribution of this study is significant. The questions it is asking are particularly interesting given the state of the literature.

Response: Thanks for your positive evaluation of our paper.

Reviewer 2 Report

I am so pleased to review this timely, very significant research as authors said that aging is one of the major challenges facing the world today.

I have enjoyed this well-written paper a lot. Please take a look at my small suggestions.

Abstract: Please add sample size and data used for analysis.

Line 14-23: Too long. Please describe shortly.

Line 45-53: need references to support these sentences.

If you add a reference of framing, that will be better. (Chong, D., & Druckman, J. N. (2007). Framing theory. Annual political science review, 10(1), 103-126.)

I think that instead of "We," "This study" will be a better way to express it.

For the variables such as Susceptibility and Benefits, I suggest authors cite health belief model. " Rosenstock, I. M. (1974). Historical origins of the health belief model. Health education monographs, 2(4), 328-335."

line 184: 3, 000, 000 -> 3,000,000

The authors showed the results of the analysis but did not mention SEM software. I assume AMOS.

Author Response

Reviewer #2:

Comment 1: Add sample size and data used for analysis in the abstract.

Response: According to reviewer’s comment, we’ve clarified the number of valid samples in the abstract (please see Line 13).

Comment 2: Describe the conclusions shortly (Line 14-23).

Response: We agreed with this comment and we’ve rewritten the conclusions in the abstract (please see Line 16-23). The revised conclusions include the main effect, the mediating effect and the moderating effect, which we believe to be more concise and organized.

Comment 3: Some references are needed to support the theoretical background (Line 45-53).

Response: We are sorry for the negligence of not attaching references and we’ve referred to some recent and highly-cited studies to explain the theoretical background.

Comment 4: Better references of framing, susceptibility and benefits should be cited.

Response: We are grateful for this suggestion and we’ve updated our reference to framing theory by citing Chong and Druckman’s (2007) work (please see Ref 34). And we introduced the variables such as susceptibility and benefits by citing Rosenstock’s (2005) work (please see Ref 28), which is a follow-up to the Health Belief Model.

Comment 5: Correct the punctuation (Line 184).

Response: We are so sorry about the misuse of punctuation in the original manuscript. As suggested by the reviewer, we’ve changed 3, 000, 000 to 3,000,000 (please see Line 192).

Comment 6: The SEM software used should be mentioned.

Response: The CFA was conducted by AMOS. Following the advice from reviewer, we’ve specified the software used when reporting the results (please see Line 272).

Thank you so much for your helpful advice.

Reviewer 3 Report

Please, see the attached PDF file.

Author Response

Reviewer #3:

Comment 1: The length of the first sentence should be reduced and the methodological part, especially the conduct of an experiment and the type of sample, should be increased in the abstract (see Line 8-24).

Response: Thank you for your suggestions, and we’ve simplified the description of the background and introduced the methodological part in a more detailed way (please see Line 8-15).

Comment 2: Update the literature used in the introduction part.

Response: Following the reviewer’s suggestions, we’ve updated our references in the introduction part as possible (please see the Reference part).

Comment 3: The statement of the objectivity of information about services is not firm enough (Line 57).

Response: We agree with the reviewer’s opinion that information about services is never objective, and we’ve also changed our expression into a firmer way (see Line 60).

Comment 4: The author should focus better on information and its framework, rather than emotions (persuasion) and messages (Line 54-75).

Response: It’s really true as the reviewer suggested that we should focus better on information framework, since it serves as the independent variable in our research. And we’ve reviewed the literature on message framing and proposed a hypothesis of its effect in section 2.2. But we still believe that it’s necessary to explain the path between information stimulus and individuals’ behavior, for the reason that making a purchase decision is usually a process of being persuaded, and it’s concerned with the mediating and moderating variables in our research. Therefore, we would like to keep this part.

Comment 5: Indicate basic references to the theories used and explain the reasons for using them (Line 80-81).

Response: As suggested by the reviewer, we’ve added the basic references to the theories used in the paper (please see Line 85; Ref 33-34). Meanwhile, we’ve also explained the role that these two theories played in our research (please see Line 85-87).

Comment 6: Since the socioemotional selectivity theory is based on motivations, the authors should explain how motivations are reflected in the model.

Response: We strongly agree that SST is a motivation-based theory, and we applied it to our research because the theory also pointed out the cognitive changes grounded in motivational shift (Carstensen & Hershfield, 2021, Ref 41). Following the reviewer’s suggestion, we’ve tried to explain the link between motivational shifts and the positivity effect of the elderly and added some references (please see Line 102-103).

Comment 7: The authors did a survey instead of an experimental design, and they should clearly explain the process of sampling and the selection of participants in a survey design (Line 181).

Response: We feel sorry that we didn’t elaborate the implementation of experiment in our paper, and according to the reviewer’s advice, we’ve added how the participants were recruited (please see Line 289-296). And we would like to say that we did adopt an experiment design in our research. Drawing on previous studies using an experiment method (Ye et al., 2021; Li et al., 2022), we manipulated the independent variable (i.e., message framing), randomly assigned subjects to different groups, and used questionnaires to collect data.

Reference:

  1. Ye, W.; Li, Q.; Yu, S. Persuasive Effects of Message Framing and Narrative Format on Promoting COVID-19 Vaccination: A Study on Chinese College Students. Int. J. Environ. Res. Public Health 2021, 18, 9485. https://doi.org/10.3390/ijerph18189485
  2. Li, Y.; Zhu, Y.; Zhang, G.; Zhou, J.; Liu, J.; Li, Z.; He, B. The Effects of Anthropomorphism, Message Framing, and Voice Type on Unhealthy Sleep Behavior in Young Users: The Mediating Role of Risk Perception. Int. J. Environ. Res. Public Health 2022, 19, 9570. https://doi.org/10.3390/ijerph19159570

Comment 8: Cronbach's alpha is meaningless if you calculate the composite reliability.

Response: Thanks for elucidating the differences of these two reliability methods. The reason why we calculated Cronbach’s alpha is that it’s the most widely used estimator of test and scale reliability in the social sciences (Peterson & Kim, 2013). But we’ve also realized its limitations and calculated the composite reliability, so as to reconfirm that our scales used have good reliability. Since the results of both indicators are good, they are both remained.

Reference:

  1. Peterson, R. A.; Kim, Y. On the Relationship between Coefficient Alpha and Composite Reliability. J. Appl. Psychol. 2013, 98 (1), 194–198. https://doi.org/10.1037/a0030767.

Comment 9: Limitations should also include the point in time at which the study was conducted and the selection of the sample.

Response: Thank you for your helpful advice, and we’ve revised the limitations of this paper accordingly (please see Line 443-445).

We deeply appreciate your useful suggestions.
